# Enzyme I facilitates reverse flux from pyruvate to phosphoenolpyruvate in *Escherichia coli*

Christopher P. Long[1],*, Jennifer Au[1],*, Nicholas R. Sandoval[1],†, Nikodimos A. Gebreselassie[1] & Maciek R. Antoniewicz[1]

The bacterial phosphoenolpyruvate-carbohydrate phosphotransferase system (PTS) consists of cascading phosphotransferases that couple the simultaneous import and phosphorylation of a variety of sugars to the glycolytic conversion of phosphoenolpyruvate (PEP) to pyruvate. As the primary route of glucose uptake in *E. coli,* the PTS plays a key role in regulating central carbon metabolism and carbon catabolite repression, and is a frequent target of metabolic engineering interventions. Here we show that Enzyme I, the terminal phosphotransferase responsible for the conversion of PEP to pyruvate, is responsible for a significant *in vivo* flux in the reverse direction (pyruvate to PEP) during both gluconeogenic and glycolytic growth. We use [13]C alanine tracers to quantify this back-flux in single and double knockouts of genes relating to PEP synthetase and PTS components. Our findings are relevant to metabolic engineering design and add to our understanding of gene-reaction connectivity in *E. coli*.

[1] Department of Chemical and Biomolecular Engineering, Metabolic Engineering and Systems Biology Laboratory, University of Delaware, Newark Delaware 19716, USA. * These authors contributed equally to this work. † Present address: Department of Chemical and Biomolecular Engineering, Tulane University, New Orleans, LA 70188, USA. Correspondence and requests for materials should be addressed to M.R.A. (email: mranton@udel.edu).

The phosphoenolpyruvate-carbohydrate phosphotransferase system (PTS) is used by many bacteria and some archaea for the uptake and phosphorylation of sugar substrates[1]. It is the main mechanism of glucose uptake and utilization in the model organism *Escherichia coli*, where it also has an important role in carbon catabolite repression and regulating central carbon metabolism[1–3]. The PTS consists of four proteins carrying out successive phosphotransferase reactions, coupling glucose transport and phosphorylation to the lower glycolytic reaction of phosphoenolpyruvate (PEP) to pyruvate (PYR). This allows for the coupled regulation of substrate uptake and glycolytic flux, as the PEP/PYR ratio has been shown to act as part of a flux sensor[4,5] and controller of phosphofructokinase (encoded by *pfkA*) activity via allosteric inhibition by PEP[6]. Owing to its central metabolic function and complex regulatory role, the PTS is a frequent target of metabolic engineering interventions[7–10].

Although individual steps of the PTS are known to be reversible[1,3,11], current understanding allows only for a net forward flux during the uptake of a PTS sugar (for example, glucose). Indeed, the conversion from PEP to PYR, which is also facilitated by pyruvate kinases (encoded by *pykA* and *pykF* in *E. coli*), is often assumed to be a committed step in lower glycolysis. This assumption has practical implications, for example in the analysis of stable isotope labelling data through $^{13}C$ metabolic flux analysis and in flux balance analysis studies. The reverse reaction, PYR to PEP, is carried out by the gluconeogenic enzyme PEP synthetase (PpsA, encoded by *ppsA* in *E. coli*). This enzyme is minimally expressed during growth on glycolytic substrates[12], as significant activity would cause a wasteful futile cycle. However, PpsA is actively expressed under gluconeogenic conditions via transcriptional regulation by Cra[13].

In this work, we show that Enzyme I (EI), the terminal phosphotransferase in the PTS responsible for the conversion of PEP to PYR, is responsible for a significant *in vivo* flux in the reverse direction (that is, PYR to PEP) during both gluconeogenic and glycolytic growth. We use knockout strains and $^{13}C$ alanine tracer experiments to directly quantify this reverse flux and determine gene–reaction relationships. We demonstrate that PpsA and EI are able to interchangeably and exclusively support the major gluconeogenic flux from PYR to PEP during growth on acetate and pyruvate. Similar experiments under growth on glycolytic substrates glucose and xylose demonstrate that this reverse flux is mainly attributable to EI, indicating an unexpected role for this enzyme in the context of central carbon metabolism. Furthermore, we show that this reverse flux is modulated by genetic perturbation of other PTS components.

## Results

**Enzyme I supports a significant gluconeogenic flux.** There are two possible gluconeogenic routes for acetate metabolism (Fig. 1a). Acetate enters central carbon metabolism as acetyl-CoA (AcCoA) and can either be metabolized to PEP via PEP carboxykinase (Pck; shown in green in Fig. 1a) or via malic enzyme (MaeAB) followed by conversion of PYR to PEP (shown in purple). As discussed above, this latter reaction is known to be carried out under gluconeogenic conditions by PEP synthetase (PpsA).

To resolve the relative contribution of these two gluconeogenic routes, a tracer experiment using [1-$^{13}C$]alanine was applied (Fig. 1). The tracer was added during growth on excess acetate (growth rates shown in Fig. 1b). Alanine equilibrates with intracellular PYR (Fig. 1f), which results in a PYR pool (observed via valine labelling) that is a mixture of unlabelled PYR (M0) produced from unlabelled sources in central carbon metabolism,

and [1-$^{13}C$]PYR (M1) produced from the tracer (Fig. 1c). As oxaloacetate (OAC, observed via aspartate) is almost entirely unlabelled (Fig. 1d; the labelled C-1 of PYR is lost in the pyruvate dehydrogenase reaction before entering the TCA cycle), the relative contribution of each route to PEP production (as measured by phenylalanine labelling, Fig. 1e) is easily calculated (Supplementary Data 1).

In the wild type, a significant amount (~60%) of PEP was generated from PYR (Fig. 1g). To confirm that PpsA was responsible for this flux, the tracer experiment was repeated with a Δ*ppsA* knockout strain. Surprisingly, the contribution of PYR to PEP (~65%) was similar to the wild type. Following a database search for enzymes able to interconvert PYR and PEP[14], we hypothesized that Enzyme I (EI, encoded by the gene *ptsI*) may be involved. EI is known to react reversibly[15], but is not known to have a role in gluconeogenesis. In the knockout strain Δ*ptsI*, the contribution of PYR to PEP (~65%) was still similarly high to the wild-type and Δ*ppsA* strains. To determine whether any other enzymes were involved with this flux, a double knockout, Δ*ppsA*Δ*ptsI*, was constructed and the tracer experiment was repeated. In this double knockout, PEP labelling was entirely eliminated, indicating that the flux from PYR to PEP was zero (Fig. 1e,g). These results suggest that PpsA and EI interchangeably and exclusively support the large gluconeogenic flux from PYR to PEP observed in the wild type during growth on acetate. This result was also observed during growth on pyruvate (Supplementary Fig. 1). With pyruvate, the WT, Δ*ppsA* and Δ*ptsI* strains grew similarly well (~0.25 h$^{-1}$), while Δ*ppsA*Δ*ptsI* was unable to grow (Supplementary Fig. 1a). For the three viable strains, nearly 100% of PEP was generated directly from PYR (Supplementary Fig. 1b), indicating that alternative routes via the glyoxylate shunt and PCK or MAE reactions were not utilized. The unaffected growth rates and PYR to PEP fluxes in both Δ*ppsA* and Δ*ptsI* during growth on acetate and pyruvate reveal a high degree of flexibility in the system, which requires either rapid transcriptional compensation or large excess capacity for each enzyme.

**A significant back-flux is measured during growth on glucose.** Given the surprising activity of EI under gluconeogenic growth conditions, we next sought to determine whether there was any measureable flux from PYR to PEP during growth on glucose. This flux was expected to be minimal or nonexistent, as this reaction is traditionally understood to have a large forward thermodynamic driving force, and gluconeogenic flux via PpsA would create a futile cycle. Glucose is a PTS sugar, meaning that during its consumption EI actively participates in the conversion of PEP to PYR. The experimental approach was modified slightly from the acetate and pyruvate cases by using [U-$^{13}C$]alanine as tracer instead of [1-$^{13}C$]alanine (Fig. 2). Again, PYR labelling was observed via valine labelling, OAC via aspartate, and PEP labelling via phenylalanine labelling. The contributions to PEP from OAC and PYR were distinguishable due to M1 labelling in OAC, generated from scrambling in the TCA cycle (Fig. 2f). During growth on glucose, the wild type was determined to have a statistically significant back-flux through which 10% of PEP was generated from PYR. When Δ*ppsA* strain was analysed, it was found to have the same back-flux as the wild type (Fig. 2g), a result consistent with the reports of minimal PpsA expression during growth on glucose[12]. Given the dual contribution of PpsA and EI to gluconeogenic flux under growth on acetate and pyruvate, it was suspected that EI may also be responsible for this back-flux on glucose.

However, the EI mutant (Δ*ptsI*) is known to grow minimally on glucose, and only after a long lag phase[10,16]. After pre-growth

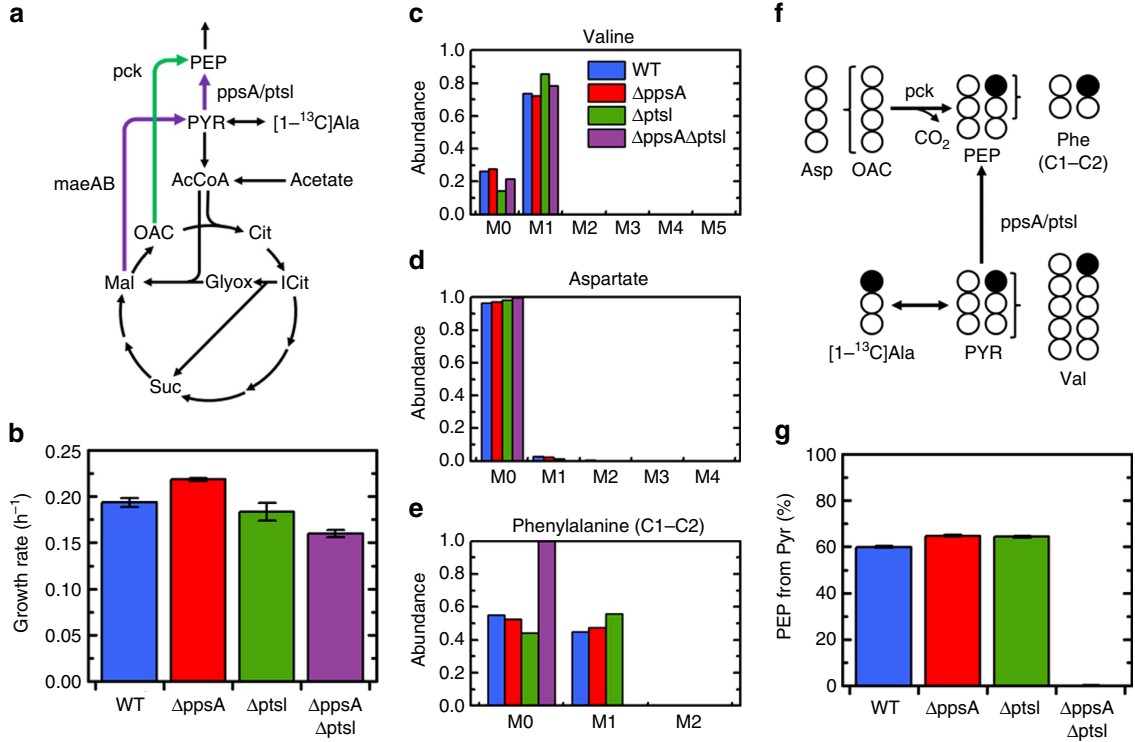

**Figure 1 | Quantification of alternative routes of PEP generation during growth on acetate.** (**a**) Schematic showing two routes of PEP synthesis during growth on acetate. After malate is produced via glyoxylate shunt, malic enzyme (*maeAB*) can convert malate to PYR, from which PEP can be formed by the activity of *ppsA* or *ptsI*. Alternatively, *pck* can convert oxaloacetate (OAC) to PEP directly. (**b**) Growth rates of four strains during growth on acetate, wild-type (WT), Δ*ppsA*, Δ*ptsI* and the double-knockout Δ*ppsA*Δ*ptsI*. (**c**) Labelling of valine from [1-$^{13}$C]alanine, reflecting PYR labelling. Labelling is M1 (from tracer) and M0 (from unlabelled precursors in central metabolism). (**d**) Labelling of aspartate from [1-$^{13}$C]alanine, reflecting OAC labelling. Aspartate is almost entirely unlabelled (M0). (**e**) Labelling of the first two (C1–C2) carbons of phenylalanine, reflecting the labelling of the first two carbons of PEP. (**f**) Schematic depicting the conversion of [1-$^{13}$C]alanine to PEP and the measured amino acids. Opened and filled circles represent unlabelled ($^{12}$C) and labelled ($^{13}$C) carbons, respectively. (**g**) Percentage of PEP generated from PYR. Approximately 60% of PEP is generated from PYR in the WT and each single knockout strain; however, the flux is completely eliminated in the double knockout, indicating dual responsibility of *ppsA* and *ptsI* for the conversion of PYR to PEP. Data presented in **b** are mean ± s.e.m. of two biological replicates. Labelling data in **c**–**e** have been corrected for natural abundances and unlabelled biomass present before tracer introduction. The error presented in **g** reflects the propagation of GC–MS measurement error through the calculation.

on LB medium and transferring to minimal medium with glucose, little or no growth was observed over 60 h (Supplementary Fig. 2a). Previous studies have found that growth on glucose can be facilitated by the induction of the *gal* operon, natively used for galactose transport and metabolism[7]. Upon induction, the GalP proton-symport transporter is able to non-specifically transport glucose[10,16–18], which can be subsequently phosphorylated by glucokinase (*glk*). To take advantage of this phenomenon, Δ*ptsI* and Δ*ppsA*Δ*ptsI* strains were pre-grown in minimal medium with galactose. Upon transferring to minimal medium with glucose, growth immediately commenced, albeit at a relatively slow growth rate (~0.1 h$^{-1}$; Fig. 2b, Supplementary Fig. 2a). The absence of PTS transport was confirmed by the lack of growth under the same conditions of a Δ*ptsI*Δ*glk* double knockout strain (Supplementary Fig. 2b).

In both the Δ*ptsI* and Δ*ppsA*Δ*ptsI* strains first pre-grown in this way, a significant amount of PEP labelling was observed (Fig. 2e). In Δ*ptsI*, 13% of PEP came from PYR, indicating significant PpsA activity (Fig. 2g), which is likely a result of active expression of *ppsA* in the altered regulatory state of this strain. In the Δ*ppsA*Δ*ptsI* strain, almost all of the PEP labelling was generated from OAC via the PCK reaction (note the significant M1 labelling in Fig. 2e). As a result, only 2.4% of PEP was generated from PYR, demonstrating again responsibility of EI and PpsA for this flux.

It is important to note that in these experiments WT and Δ*ppsA* strains take up glucose via the PTS system, while Δ*ptsI* and Δ*ppsA*Δ*ptsI* strains are using non-PTS transporters. To get a more direct comparison of these four strains during glycolytic growth, the non-PTS sugar xylose was chosen for subsequent experiments.

**Enzyme I accounts for back-flux during growth on xylose.** The [U-$^{13}$C]alanine tracer experiments were repeated for the same four strains (WT, Δ*ppsA*, Δ*ptsI* and Δ*ppsA*Δ*ptsI*) using the non-PTS sugar xylose as substrate (Fig. 3). Xylose is transported into the cell via an ABC transporter (*xylGHF*) or a proton symporter (*xylE*)[19] (Fig. 3a), which renders EI non-essential for growth (Fig. 3b). During growth on xylose, the back-flux observed in the wild type was similar to that observed during growth on glucose, with 11% of PEP formed from PYR (Fig. 3g). Once again, this flux was not significantly reduced in the Δ*ppsA* strain. However, this flux was almost completely eliminated in the Δ*ptsI* and Δ*ppsA*Δ*ptsI* strains (Fig. 3g), providing strong evidence that EI was exclusively responsible for the conversion of PYR to PEP under these conditions. Thus, during growth on xylose, PpsA was not active as may be expected under normal glycolytic growth conditions[12].

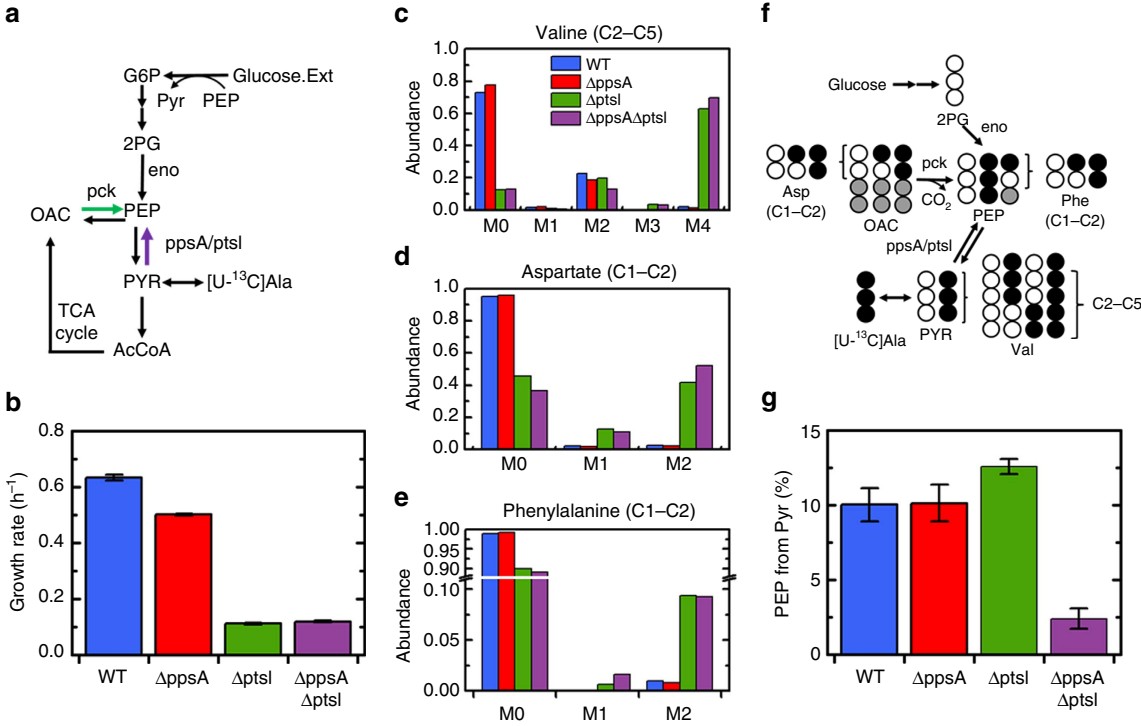

**Figure 2 | During growth on glucose there is a significant back-flux from PYR to PEP not carried out by PEP synthetase (ppsA).** (**a**) Schematic of glucose consumption and metabolism related to PEP and PYR interconversion. Glucose is transported and phosphorylated by the PTS, simultaneously converting PEP to PYR via Enzyme I (ptsI). (**b**) Growth rates of four strains during growth on glucose; $\Delta ptsI$ and $\Delta ppsA\Delta ptsI$ were pre-grown on galactose to facilitate growth on glucose. (**c**) Labelling of last four carbons (C2–C5) of valine, representing the condensation of the last two carbons (C2–C3) of two PYR molecules. Labelling is mainly M0 (condensation of two unlabelled PYRs), M2 (condensation of one fully labelled PYR and one unlabelled PYR) and M4 (condensation of two labelled PYRs). (**d**) Labelling of the first two (C1–C2) carbons of aspartate, reflecting the labelling of the same carbons in OAC. M1 labelling is generated through scrambling in the TCA cycle. (**e**) Labelling of the first two (C1–C2) carbons of phenylalanine, reflecting the labelling of the first two carbons of PEP. (**f**) Schematic depicting the conversion of [U-13C]alanine to PEP and the measured amino acids. The relative contributions of the three sources of PEP were quantified via regression. Opened and filled circles represent unlabelled (12C) and labelled (13C) carbons, respectively. (**g**) Percentage of PEP generated from PYR. Approximately 10% is generated from PYR in both WT and $\Delta ppsA$ strains. The contribution is slightly elevated in $\Delta ptsI$, likely due to activity of ppsA. There is minimal back-flux in $\Delta ppsA\Delta ptsI$. Data presented in **b** are mean ± s.e.m. of two biological replicates. Labelling data in **c**–**e** have been corrected for natural abundances and unlabelled biomass present before tracer introduction. The error presented in **g** reflects the propagation of GC-MS measurement error through the calculation.

**Back-flux is affected by genetic knockouts of PTS components.** Given the strong evidence for EI involvement in the back-flux from PYR to PEP under glycolytic conditions, it was further hypothesized that this activity would be perturbed in knockout mutants of other PTS components. The PTS and its components are shown in Fig. 4a. The phosphotransferase partner of EI is HPr (encoded by the gene ptsH), which then interacts with the soluble (crr) and membrane-bound (ptsG) components of the glucose-specific Enzyme II complex (EIIABC$^{Glc}$). The growth rates on glucose and xylose for the mutant strains $\Delta ptsG$, $\Delta crr$ and $\Delta ptsH$ are shown in Fig. 4b. The mannose Enzyme II complex (EII$^{Man}$) is also known to transport glucose[20], allowing for the $\Delta ptsG$ and $\Delta crr$ strains to grow on glucose with no lag phase, presumably still utilizing the PTS system. The lack of significant lag phase or growth defect for $\Delta ptsH$ on glucose is less clear, but it has been suggested that this strain may be able to recruit non-PTS transporters such as GalP more quickly than $\Delta ptsI$[16]. Alternatively, it has been suggested that the HPr-like protein FPr from the fructose PTS may be able to substitute its activity for HPr[21].

The [U-13C]alanine tracer experiments described above for glucose and xylose were performed for all knockouts of PTS components. There was a striking increase in the back-flux for several knockout strains (Fig. 4c). For example, in the $\Delta ptsG$

strain grown on glucose, 21% of PEP was formed from PYR. Similarly high back-fluxes were also observed for $\Delta crr$ on both glucose (26%) and xylose (24%). However, the back-flux was nearly eliminated in $\Delta ptsH$ on glucose, and significantly reduced on xylose. This indicates that HPr is likely the primary, if not sole, phosphotransferase partner of EI.

Although PpsA is generally not expected to be expressed during normal glycolytic conditions, it could be expressed and active in the altered regulatory environments of these mutant strains, as was seen in the case of $\Delta ptsI$ on glucose. The phosphorylation states of PTS proteins are known effectors in regulatory circuits, particularly in the signalling of glucose availability for global metabolic regulation[3,22]. Perturbations in the PTS system could conceivably result in expression of gluconeogenic genes, for example, through the activation of the global regulator Cra. To help to assign responsibility for the increased back-flux in these strains, several double knockouts of ptsG, crr, ppsA and ptsI were constructed (Fig. 4b,c). The strain $\Delta ptsG\Delta ppsA$ had a significant reduction in back-flux, down to 10% from 21% in the single knockout during growth on glucose (Fig. 4c). Thus the excess back-flux in $\Delta ptsG$, relative to the wild-type, was attributable to PpsA. In contrast, there was only a modest decrease in back-flux for $\Delta crr\Delta ppsA$ on glucose, and there was no decrease on xylose. In the $\Delta crr\Delta ptsI$ strain, pre-grown on

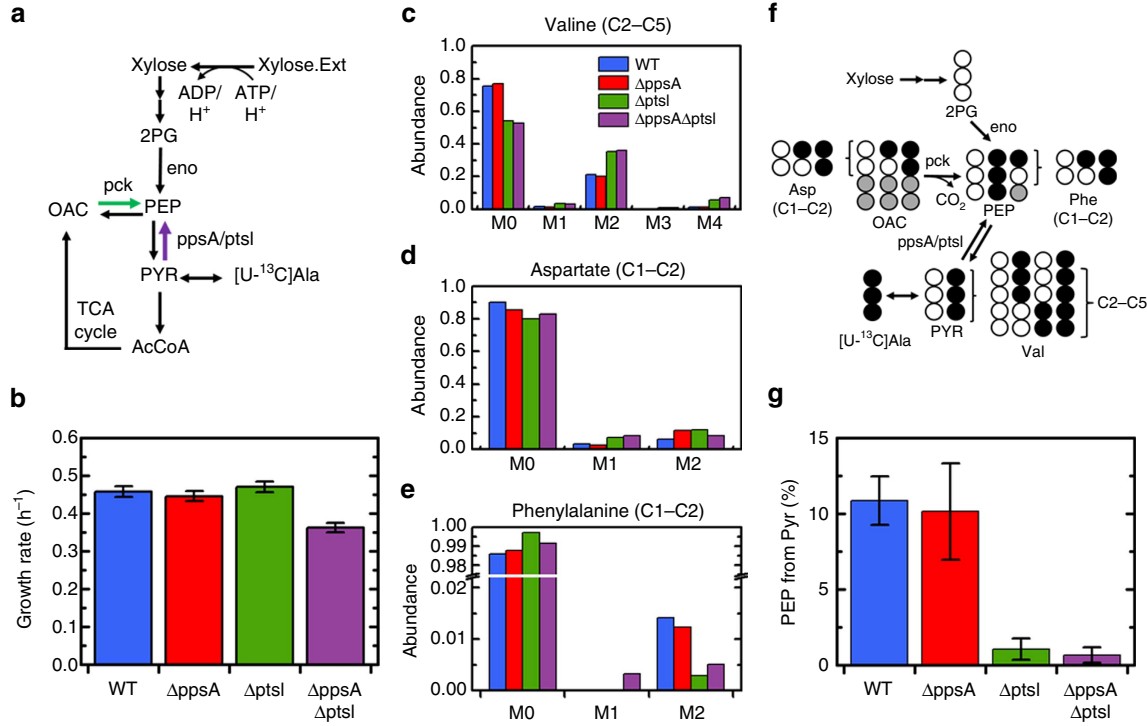

**Figure 3 | Enzyme I (*ptsI*) is responsible for the back-flux from PYR to PEP during growth on xylose.** (**a**) Schematic of xylose consumption and metabolism related to PEP and PYR interconversion. Xylose is transported via non-PTS transporters. (**b**) Growth rates of four strains during growth on xylose, wild-type (WT), Δ*ppsA*, Δ*ptsI* and the double-knockout Δ*ppsA*Δ*ptsI*. (**c**) Labelling of last four carbons (C2–C5) of valine, representing the condensation of the last two carbons (C2–C3) of two PYR molecules. Labelling is mainly M0 (condensation of two unlabelled PYRs), M2 (condensation of one fully labelled PYR and one unlabelled PYR) and M4 (condensation of two labelled PYRs). (**d**) Labelling of the first two (C1–C2) carbons of aspartate, reflecting the labelling of the same carbons in OAC. M1 labelling is generated through scrambling in the TCA cycle. (**e**) Labelling of the first two (C1–C2) carbons of phenylalanine, reflecting the labelling of the first two carbons of PEP. (**f**) Schematic depicting the conversion of [U-$^{13}$C]alanine to PEP and the measured amino acids. The relative contributions of the three sources of PEP were quantified via regression. Opened and filled circles represent unlabelled ($^{12}$C) and labelled ($^{13}$C) carbons, respectively. (**g**) Percentage of PEP generated from PYR. Approximately 10% is generated from PYR in both the WT and Δ*ppsA* strains. The flux is nearly completely eliminated in the Δ*ptsI* and Δ*ppsA*Δ*ptsI* strains, indicating a major role for Enzyme I (*ptsI*) in facilitating the back-flux. Data presented in **b** are mean ± s.e.m. of two biological replicates. Labelling data in **c**–**e** have been corrected for natural abundances and unlabelled biomass present before tracer introduction. The error presented in **g** reflects the propagation of GC-MS measurement error through the calculation.

galactose as previously described, there was a reduction in the back-flux during growth on glucose, down to 14% from 26% in Δ*crr*. On xylose, the flux for the double knockout was nearly zero, compared with 24% in the single knockout. The fact that Δ*crr*Δ*ptsI* on glucose still had some back flux likely indicates some degree of PpsA activity. Therefore, both enzymes play a role in the elevated back-flux of Δ*crr* on glucose. During growth on xylose, however, the elevated back-flux is entirely the result of EI.

Overall, the double-knockout studies are helpful in deconvoluting the relative contributions of PpsA and EI. These results suggest that in certain PTS mutant strains, particularly Δ*ptsI*, Δ*crr* and Δ*ptsG*, the perturbed regulatory environments during growth on glucose results in PpsA being expressed and active. PpsA does not appear to play a role during growth on xylose, allowing for a more direct analysis of the interactions of PTS component perturbations with the back-flux activity.

## Discussion

The results presented here show that Enzyme I facilitates reverse flux from PYR to PEP in *E. coli*. This function is active both under conditions in which the PTS is the primary means of transporting substrate (growth on glucose) and is not (growth on acetate, pyruvate or xylose). Knowledge of this gene–reaction

relationship will improve our understanding and annotation of *E. coli* central carbon metabolism, which is of central importance in metabolic modelling and engineering efforts such as $^{13}$C metabolic flux analysis[23] and the development of production strains[7–10].

This metabolic activity also raises biological questions about the PTS, its regulation and whether there are additional unannotated connectivities in the network. For example, are there other kinases that interact with HPr or EI to provide the phosphoryl groups needed to sustain this large flux from PYR to PEP? In the gluconeogenic conditions studied here, there is clearly an unknown phosphate donor that allows EI to drive a large net flux from PYR to PEP. In the glycolytic conditions, it is less clear whether the observed labelling is a result of simple reversibility in the EI/HPr system, or whether there is another donor specifically responsible for the PYR to PEP flux. The large increase in the back-flux in Δ*crr*, along with its elimination in Δ*ptsH* during growth on glucose, is consistent with a model in which HPr is the primary or sole phosphotransferase partner of EI, and the Δ*crr* mutation perturbs the equilibria of the PTS chain as the less abundant or efficient EII$^{Man}$ is substituted for EII$^{Glc}$. This would cause the phosphorylated form of HPr to accumulate, driving the partial reversal of the EI and PEP/PYR reactions. The fact that this also occurs during growth on xylose is surprising,

and indicates either a robust dynamic equilibrium in the PTS system even when not being actively used or the activity of other unknown factors.

The complexity of the enzymatic and regulatory interactions in the PTS system require caution when interpreting these results. It is possible that EI could phosphorylate or activate another, yet unknown enzyme which phosphorylates PYR, or that the altered phosphorylation states of PTS components in the mutant strains studied result in the activation of such an enzyme. For example, $P \sim EIIA^{Glc}$ activates adenylate cyclase, generating cAMP that

activates the global regulator Crp[3], which controls transcription of over 100 genes[24]. Other regulatory functions spanning from carbon and nitrogen metabolism to chemotaxis are directly influenced by the phosphorylation state of PTS components[1]. In this work, we were able to deconvolute the role of PpsA, which was active in some knockout strains. Further work is needed to clarify the network, including potential candidates implicated in this work (for example, a phosphotransferase donor during gluconeogenic growth) as well as to rule out the presence of any more unknown intermediary enzymes.

The reversibility of the PEP to PYR step in glycolysis is also surprising in the global sense, particularly in the context of textbook understanding of this reaction as a 'committed step' with a large Gibbs free energy drop. In fact, the results presented here are consistent with a recent study that estimated *in vivo* $\Delta G$ values in central carbon metabolism using measured metabolite concentrations and observed cellular $\Delta G$ values[25]. For this reaction, the estimated $\Delta G$ was significantly lower than historically assumed. Similar labelling studies to those presented here demonstrated the reversibility of pyruvate kinase in human iBMK cells[25]. Taken together, these results are cause for reconsideration of our understanding of the thermodynamics, control, and engineering targets of central carbon metabolism.

## Methods

**Materials.** Chemicals and culture media were purchased from Sigma-Aldrich (St Louis, MO, USA). [1-13C]Alanine (99 atom% 13C) and [U-13C]alanine (98 + atom% 13C) were purchased from Cambridge Isotope Laboratories (Andover, MA, USA). M9 minimal medium was used for all the labelling experiments. All the solutions were sterilized by filtration.

**Strains.** For wild-type *E. coli* grown on acetate, *E. coli* K-12 MG1655 (ATCC Cat. No. 700925, Manassas, VA, USA) was used. For all other cultures, *E. coli* strains were obtained from the Keio collection (GE Dharmacon), which was generated by one-step inactivation of all non-essential genes in *E. coli* K-12 BW25113 ($\Delta(araD-araB)567$, $\Delta(lacZ4787::rrnB-3)$, lambda $-$, rph-1, $\Delta(rhaD - rhaB)568$, hsdR514; ref. 26). The strains used in this study, with identifying information from the collection, are listed in Supplementary Table 1. Double-deletion strains were constructed following the method of Datsenko and Wanner on existing Keio collection strains[26,27]. Kanamycin resistance cassettes were cured by transformation of pCP20 (ref. 28), which carries the FLP recombinase gene; the pCP20 plasmid was subsequently cured by growth at 42 °C overnight and confirmed via replica plating and PCR amplification. Kanamycin resistance cassettes for second-gene knockouts were amplified from Keio collection single deletion strains using the original Keio collection primers with homologous regions corresponding to the desired deletion. The purified amplicon was electroporated into the cured single deletion host *E. coli* strain expressing 1 mM arabinose-induced λ-Red recombinase genes from the pKD46 plasmid[27] and grown on solid LB with

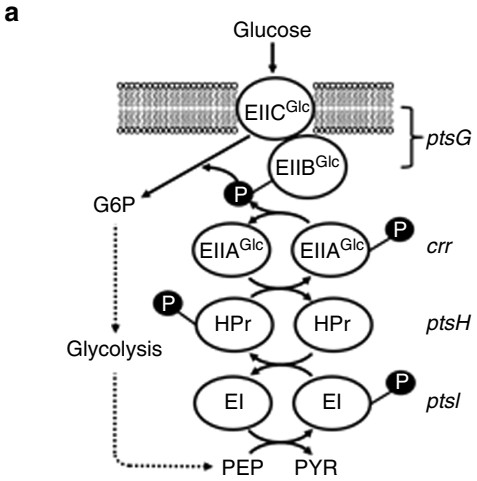

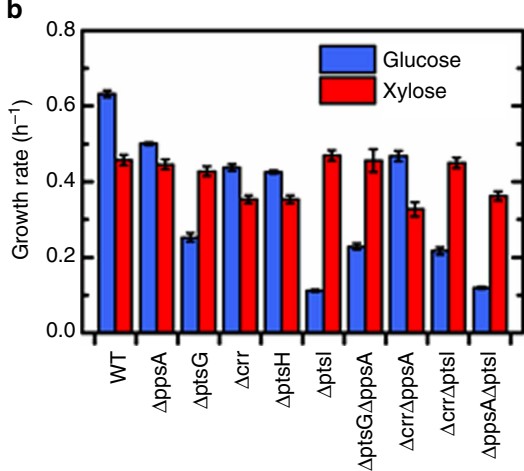

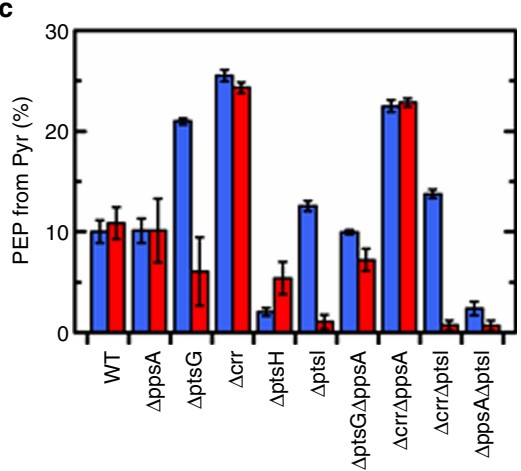

**Figure 4 | Genetic perturbations of PTS components significantly impact the back-flux from PYR to PEP.** (**a**) Schematic of the PTS sugar transport system, which couples the transport and phosphorylation of glucose at EIIBC$^{Glc}$ (*ptsG*) to conversion of PEP to PYR (*ptsI*), via the phosphotransferases ptsH (*ptsH*) and EIIA$^{Glc}$ (*crr*). (**b**) Growth rates of wild-type (WT) *E. coli* and nine knockout strains, including all single knockouts of PTS components and selected double knockouts, grown on glucose and xylose. All *ptsI* knockout strains grown on glucose were pre-grown on galactose. (**c**) [U-13C]alanine experiments were performed for all strains, and the percentage of PEP derived from PYR was determined. Several strains had significantly higher percentages of PEP derived from PYR, particularly $\Delta ptsG$ on glucose and $\Delta crr$ on glucose and xylose, indicating that PTS component perturbation impacts back-flux. Double knockouts of these strains and $\Delta ppsA$ or $\Delta ptsI$ showed that *ppsA* plays a significant role during growth on glucose, accounting for all of the elevated back-flux in $\Delta ptsG$, and for some in $\Delta crr$, as evidenced by the residual flux in $\Delta crr\Delta ptsI$. On xylose, the elevated back-flux in $\Delta crr$ is caused exclusively by EI. Data presented in **b** are mean ± s.e.m. of two biological replicates. The error presented in **c** reflects the propagation of GC-MS measurement error through the calculation.

kanamycin at 37 °C. Successful recombination was confirmed via PCR of both the mutated loci. The recombination plasmid pKD46 was subsequently cured by growth overnight at 42 °C and confirmed via replica plating. All strains carrying pCP20 and pKD46 were grown at 30 °C.

**Culture conditions.** *E. coli* strains were cultured aerobically in M9 minimal medium at 37 °C in aerated mini-bioreactors with 10 ml working volume[29]. Cultures were inoculated at $A_{600}$ of 0.01, and biomass concentration and growth rates were determined by periodic measurements of $D_{600}$ using a spectrophotometer (Eppendorf BioPhotometer). The medium contained, for the respective experiments, $1.2\,g\,l^{-1}$ acetate, $2\,g\,l^{-1}$ pyruvate, $2\,g\,l^{-1}$ glucose, or $4\,g\,l^{-1}$ xylose. In the acetate and pyruvate experiments, a bolus of 1 mM [1-$^{13}$C]alanine was added when the cultures reached a $D_{600}$ of $\sim$0.1, and the cells were collected for analysis at a $D_{600}$ of 0.5. In the glucose and xylose experiments, a bolus of 10 mM [U-$^{13}$C]alanine was added when the culture reached a $D_{600}$ of 0.5 and the cells were collected at a $D_{600}$ of 1.5. In all the experiments, the non-tracer substrate (that is, acetate, glucose or xylose) was not limiting throughout, and exponential growth was maintained (that is, culture performance was not affected by the presence of the tracer). For glucose experiments with strains containing the Δ*ptsI* mutation, pre-cultures were grown in medium with $4\,g\,l^{-1}$ galactose. The cells were centrifuged, washed and inoculated into the standard glucose medium described above at a $D_{600}$ of 0.05–0.1, and collected at a $D_{600}$ of 0.3.

**Gas chromatography mass spectrometry.** GC-MS analysis was performed on an Agilent 7890B GC system equipped with a DB-5MS capillary column (30 m, 0.25 mm inner diameter, 0.25 μm-phase thickness; Agilent J&W Scientific), connected to an Agilent 5977A Mass Spectrometer operating under ionization by electron impact (EI) at 70 eV. Helium flow was maintained at $1\,ml\,min^{-1}$. The source temperature was maintained at 230 °C, the MS quad temperature at 150 °C, the interface temperature at 280 °C and the inlet temperature at 250 °C. GC-MS analysis of tert-butyldimethylsilyl (TBDMS) derivatized proteinogenic acids was performed to measure isotopic labelling[30]. Mass isotopomer distributions were obtained by integration[31] and corrected for natural isotope abundances[32].

**Calculations.** For the [1-$^{13}$C]alanine tracer experiments, the fraction of PEP derived from PYR was determined from the labelling of PEP (determined from phenylalanine *m/z* 302 fragment labelling, C1–C2) and PYR (determined from valine *m/z* 288 fragment labelling, C1–C5):

$$\% \text{ PEP from Pyr} = \frac{\text{PEP}_{M1}}{\text{Pyr}_{M1}} = \frac{\text{Phe302}_{M1}}{\text{Val288}_{M1}} \qquad (1)$$

For the [U-$^{13}$C]alanine tracer experiments, the fraction of PEP derived from PYR was determined by least-squares regression using the measured mass isotopomer distributions (MID) of PEP (determined from phenylalanine *m/z* 302 fragment, C1–C2), OAC (determined from aspartate *m/z* 302 fragment, C1–C2), and PYR (determined from valine *m/z* 260 fragment, C2–C5), after correction for unlabelled biomass (Supplementary Methods).

$$\begin{aligned} \text{MID}_{\text{PEP}} = {}& (\% \text{ PEP from Pyr}) * \text{MID}_{\text{Pyr}} \\ & + (\% \text{ PEP from OAC}) * \text{MID}_{\text{OAC}} + (\% \text{ PEP from gluc or xyl}) * \text{MID}_{\text{unlabelled}} \end{aligned}$$
$$(2)$$

**Data availability.** All data generated or analysed during this study are included in this published article and its Supplementary Information files.

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

## Acknowledgements

This work was supported by NSF MCB-1616332 grant. C.P.L. was also supported by the University of Delaware Graduate Fellows Award.

## Author contributions

C.P.L., J.A. and N.A.G. performed cell culture and [13]C tracer experiments. N.R.S. constructed the *E. coli* knockout strains. C.P.L., J.A. and M.R.A. designed the research and wrote the paper with help from all the authors.

## Additional information

**Competing financial interests:** The authors declare no competing financial interests.

