## [Peer Review File · Nature Communications]

Reviewers' comments:

Reviewer #1 (Remarks to the Author):

Comments on the manuscript of Christopher P Long et al. to be considered for publication in Nature Communications

A) Summary: The authors have found that *E. coli* cells show a reverse flux from pyruvate to phosphoenolpyruvate (PEP) in the absence of a *ppsA* gene. From genetic and labelling studies performed under different growth conditions (glycolytic and gluconeogenic C sources), they infer that Enzyme I of the PTS is involved in this remarkable feature.

Their genetic work is based on single (*ppsA* or *ptsI*) and double mutant strains (*ppsA ptsI*) of *E. coli*. The observed growth behavior indicates that -while the double mutant is defective on glucose (as expected) - it only shows minor impediment on acetate or xylose. From the labeling experiments with either 1-¹³C-alanine or U-¹³C-alanine, the authors find label in valine as expected (two copies of pyruvate) but also label in the side chain of phenylalanine (L-phe) which they take as proof that pyruvate has been phosphorylated to PEP indeed. There is no label in L-phe, however, in the double mutant when grown on acetate. This finding could be best explained by Enzyme I being necessary for phosphorylation of pyruvate (reverse direction), especially when *ppsA* is lacking.

B) Originality and interest: This work, indeed, constitutes an important novel finding and will certainly influence the field of research, especially flux balance analysis as part of systems biology in *E. coli*.

C) The genetic approach is valid, the wealth and quality of data provided (mass isotopomer data) is convincing and well prepared. The data are adequately presented.

D) Statistics are appropriately used and error margins are given.

E): Conclusions: Purified Enzyme I~P is known to perform a reversible reaction with pyruvate (Weigel et al. 1982). For unknown reasons, this - to my knowledge- has not been taken into account by modelling approaches. Long et al. now provide experimental evidence for the in vivo reverse flux in glycolysis/gluconeogenesis at the PEP/pyruvate interface.

There is one caveat, however: the influence of Enzyme I could be regulatory instead, i.e. Enzyme I could stimulate/activate another (to be identified) enzyme which performs the actual phosphorylation step on pyruvate. Moreover, the loss of Enzyme I function affects the phosphorylation state of other proteins (HPr, Enzyme IIAGlc, CheA, acetate kinase) which in turn could be involved directly or indirectly in activation of such a pyruvate phosphorylating enzyme. For instance, EIIAGlc~P is known to activate adenylate cyclase, cAMP is then an effector together with CRP and controls more than 200 genes in *E. coli*. The authors cannot rule out this possibility and thus should discuss this.

F) Suggested improvements and questions to be addressed:

1. Line 92: The quotation of the reversible action of EI~P should not be a general PTS review (Deutscher) but attributed to the actual results of Nancy Weigel (Roseman group) JBC 1982.

2. How is the growth behavior on lactate or pyruvate of *ppsA* or *ppsA/ptsI* mutant strains? Has this been tested, and if so, what was the outcome?

Minor criticisms:

Line 21: phosphotransferases

Line 121: Xylose can also be transported into *E. coli* cells via a proton-coupled symporter, XylE. Please correct.

Line 158: PpsA (enzyme) instead of *ppsA* (gene).

Line 205: Please give proper quotation for plasmid pCP20 (Cherepanov & Wackernagel 1995, not Datsenko & Wanner 2000!)

Line 224/225: "the non-tracer substrate ... was in excess throughout": If 2g/l glucose were used this would make 11mM, whereas 1+10mM of alanine were used (lines 221, line 223), thus there was no molar excess! Please correct.

G): References: See above (F) for corrections/additions in refs.

H): Clarity and context: The summary should be rewritten to highlight the actual method used and

the proof for PEP stemming from pyruvate. In the results section, the missing label in oxaloacetate (observed via aspartate) has to be explained in more detail also taking into account possible label scrambling at the fumarate/succinate step.

Reviewer #2 (Remarks to the Author):

Manuscript: "Enzyme I facilitates reverse flux from pyruvate to phosphoenolpyruvate in *E. coli*" by Long et al.

This paper reports the role of Enzyme I (EI), the terminal phosphotransferase in the PTS and responsible for the conversion of PEP to PYR, in the significant *in vivo* reverse C flux in the direction of PYR→PEP under both gluconeogenic and glycolytic growth in *E. coli*.

Authors obtained knockout derivative mutants on this enzyme and used labeled alanine tracer experiments to quantify this reverse flux and determine gene-reaction relationships.

The authors demonstrated that this flux is a major contributor to gluconeogenesis during growth on acetate as sole carbon source, and it is supported interchangeably by both PEP synthetase (*ppsA*) and Enzyme I (*ptsI*). Similar experiments on glycolytic substrates, glucose and xylose demonstrate that this reverse flux is mainly attributable to Enzyme I. Results provide strong evidence on the role of EI in the backflow from PYR→PEP, suggesting the existence of unknown regulatory networks between EI and other potential kinases interacting with this cytoplasmic PTS component. These results are relevant for researchers in the respective field. The manuscript is well redacted, is easy to read, and some minor comments are listed below.

However, some primary concerns rely on the experimental design particularly on the EI mutant grown in xylose as non-PTS sugar to explore the back flux from PYR→PEP under glycolytic conditions in both $\Delta ptsI$ and $\Delta ptsI\Delta ppsA$ double mutants, as authors stated that EI mutant was not able to grow on glucose.

Liang et al. *Scientific Reports* 5, Article number: 13200 (2015), doi:10.1038/srep13200 reported the effect of individual knockouts of each component of the PTS:GLC in *E. coli* W3110, including, of course, the inactivation of EI, which grown on glucose as the sole carbon source. These authors reported that EI mutant (W3110I) was able to grow on glucose with a long lag phase (39 h) but interestingly achieving a higher (43 %) biomass observed respect wild-type strain and the other PTS mutants. These authors associated (but not demonstrated) the observed growth to the recruitment of GalP system.

Additionally, other reports on the characterization of derivative PTS- *E. coli* strains lacking the *pts* operon (*ptsHIcr*), have shown that these PTS- derivatives possess a reduced capability to grow in glucose as sole carbon source (e.g. Flores et al., *Nature Biotechnology* 1996;14:620-623; Flores et al. *Metabolic Engineering* 2005; 7: 70-87). These previous reports were not considered in the present submission.

In lines 121-122, authors stated that during growth on xylose, EI mutation was non-essential for growth as in glucose condition. Additionally, in lines 138-139, was wrote that "The only non-growth phenotype observed was for $\Delta ptsI$ on glucose, as previously discussed." However, according to the previous reports, particularly data from Ling et al., EI mutants can grow in glucose at low growth rates. Based on this information, authors are prompted to consider additional experiments in glucose as carbon source. Results on xylose are valuable, of course, but demonstrate the role of the inactivation of EI in the back-flow of C from PYR→PEP growing in a non-PTS sugar.

Regarding the experimental design with labeled alanine, authors are prompted to show the fate of the incorporation of this amino acid in the schematic representation of PEP synthesis in Figures 1-3 panel a, respectively.

As labeled alanine is incorporated to PEP available for cell growth, how much of the C from alanine contributed to the observed biomass in each experiment (acetate, glucose, and xylose)? Growth rates showed in Figure 1-3 panel b and Figure 4, panel b, were determined before the addition of the labeled bolus?

Minor comments:

Lines 22 and 41: Define the abbreviated form for pyruvate (PYR) as it is used in this from in the entire document. e.g. line 42: "PEP/Pyr". Use PYR instead pyruvate among the entire document.

Lines 221, show the carbon sources and concentrations assayed. Why do you use different levels of each carbon source? Does this difference be related to the total moles of each C source? Moreover, differences influenced the total biomass obtained in each carbon source? Precise.

Reviewer #1 (Remarks to the Author):

Comments on the manuscript of Christopher P Long et al. to be considered for publication in Nature Communications

A) Summary: The authors have found that *E. coli* cells show a reverse flux from pyruvate to phosphoenolpyruvate (PEP) in the absence of a *ppsA* gene. From genetic and labelling studies performed under different growth conditions (glycolytic and gluconeogenic C sources), they infer that Enzyme I of the PTS is involved in this remarkable feature.

Their genetic work is based on single (*ppsA* or *ptsI*) and double mutant strains (*ppsA ptsI*) of *E. coli*. The observed growth behavior indicates that -while the double mutant is defective on glucose (as expected) - it only shows minor impediment on acetate or xylose. From the labeling experiments with either 1-¹³C-alanine or U-¹³C-alanine, the authors find label in valine as expected (two copies of pyruvate) but also label in the side chain of phenylalanine (L-phe) which they take as proof that pyruvate has been phosphorylated to PEP indeed. There is no label in L-phe, however, in the double mutant when grown on acetate. This finding could be best explained by Enzyme I being necessary for phosphorylation of pyruvate (reverse direction), especially when *ppsA* is lacking.

B) Originality and interest: This work, indeed, constitutes an important novel finding and will certainly influence the field of research, especially flux balance analysis as part of systems biology in *E. coli*.

We thank the reviewer for the positive comments.

C) The genetic approach is valid, the wealth and quality of data provided (mass isotopomer data) is convincing and well prepared. The data are adequately presented.

We thank the reviewer for the positive comments.

D) Statistics are appropriately used and error margins are given.

E): Conclusions: Purified Enzyme I~P is known to perform a reversible reaction with pyruvate (Weigel et al. 1982). For unknown reasons, this - to my knowledge- has not been taken into account by modelling approaches. Long et al. now provide experimental evidence for the in vivo reverse flux in glycolysis/gluconeogenesis at the PEP/pyruvate interface.

There is one caveat, however: the influence of Enzyme I could be regulatory instead, i.e. Enzyme I could stimulate/activate another (to be identified) enzyme which performs the actual phosphorylation step on pyruvate. Moreover, the loss of Enzyme I function affects the phosphorylation state of other proteins (HPr, Enzyme IIAGlc, CheA, acetate kinase) which in turn could be involved directly or indirectly in activation of such a pyruvate phosphorylating enzyme. For instance, EIIAGlc~P is known to activate adenylate cyclase, cAMP is then an effector together with CRP and controls more than 200 genes in *E. coli*. The authors cannot rule out this possibility and thus should discuss this.

We thank the reviewer for the good suggestion. We have added additional discussion to this effect. Furthermore, we also discuss how such regulatory interactions are a likely the cause for *ppsA* contributions to the PYR to PEP flux in certain PTS mutant strains growing on glucose.

F) Suggested improvements and questions to be addressed:

1. Line 92: The quotation of the reversible action of EI~P should not be a general PTS review (Deutscher) but attributed to the actual results of Nancy Weigel (Roseman group) JBC 1982.

We have changed the citation as suggested.

2. How is the growth behavior on lactate or pyruvate of ppsA or ppsA/ptsI mutant strains? Has this been tested, and if so, what was the outcome?

We thank the reviewer the good suggestion. We have performed additional labeling experiments (using pyruvate as the substrate) with the WT, Δ ppsA, Δ ptsI, and Δ ppsA Δ ptsI strains. The results are discussed in the updated manuscript. Overall, the pyruvate results are in good agreement with the acetate results. With pyruvate as substrate, WT, Δ ppsA, and Δ ptsI strains grew similarly well, while Δ ppsA Δ ptsI was unable to grow. For the three viable strains, 100% of PEP was generated directly from PYR, indicating that alternative routes via the glyoxylate shunt and PCK or MAE were not utilized. The absence of a growth defect in either Δ ppsA or Δ ptsI as the PYR to PEP flux remains unchanged during growth on acetate and pyruvate shows a high degree of flexibility in the system.

Minor criticisms:

Line 21: phosphotransferases

Line 121: Xylose can also be transported into E. coli cells via a proton-coupled symporter, XylE. Please correct.

Line 158: PpsA (enzyme) instead of ppsA (gene).

Line 205: Please give proper quotation for plasmid pCP20 (Cherepanov & Wackernagel 1995, not Datsenko & Wanner 2000!)

Line 224/225: "the non-tracer substrate ... was in excess throughout": If 2g/l glucose were used this would make 11mM, whereas 1+10mM of alanine were used (lines 221, line 223), thus there was no molar excess!

Please correct.

G): References: See above (F) for corrections/additions in refs.

Corrected.

H): Clarity and context: The summary should be rewritten to highlight the actual method used and the proof for PEP stemming from pyruvate. In the results section, the missing label in oxaloacetate (observed via aspartate) has to be explained in more detail also taking into account possible label scrambling at the fumarate/succinate step.

We have updated the abstract. In the main text we also provide more details regarding how the contribution of oxaloacetate to PEP was taken into account, including scrambling of labeling in the TCA cycle which produces M1 labeled oxaloacetate.

Reviewer #2 (Remarks to the Author):

Manuscript: "Enzyme I facilitates reverse flux from pyruvate to phosphoenolpyruvate in *E. coli*" by Long et al.

This paper reports the role of Enzyme I (EI), the terminal phosphotransferase in the PTS and responsible for the conversion of PEP to PYR, in the significant in vivo reverse C flux in the direction of PYR→PEP under both gluconeogenic and glycolytic growth in *E. coli*.

Authors obtained knockout derivative mutants on this enzyme and used labeled alanine tracer experiments to quantify this reverse flux and determine gene-reaction relationships.

The authors demonstrated that this flux is a major contributor to gluconeogenesis during growth on acetate as sole carbon source, and it is supported interchangeably by both PEP synthetase (*ppsA*) and Enzyme I (*ptsI*). Similar experiments on glycolytic substrates, glucose and xylose demonstrate that this reverse flux is mainly attributable to Enzyme I.

Results provide strong evidence on the role of EI in the backflow from PYR→PEP, suggesting the existence of unknown regulatory networks between EI and other potential kinases interacting with this cytoplasmic PTS component. These results are relevant for researchers in the respective field. The manuscript is well redacted, is easy to read, and some minor comments are listed below.

We thank the reviewer for the positive comments.

However, some primary concerns rely on the experimental design particularly on the EI mutant grown in xylose as non-PTS sugar to explore the back flux from PYR→PEP under glycolytic conditions in both $\Delta ptsI$ and $\Delta ptsI \Delta ppsA$ double mutants, as authors stated that EI mutant was not able to grow on glucose.

Liang et al. Scientific Reports 5, Article number: 13200 (2015), doi:10.1038/srep13200 reported the effect of individual knockouts of each component of the PTS:GLC in *E. coli* W3110, including, of course, the inactivation of EI, which grown on glucose as the sole carbon source. These authors reported that EI mutant (W3110I) was able to grow on glucose with a long lag phase (39 h) but interestingly achieving a higher (43 %) biomass observed respect wild-type strain and the other PTS mutants. These authors associated (but not demonstrated) the observed growth to the recruitment of GalP system.

Additionally, other reports on the characterization of derivate PTS- *E. coli* strains lacking the *pts* operon (*ptsHICrr*), have shown that these PTS- derivatives possess a reduced capability to grow in glucose as sole carbon source (e.g. Flores et al., Nature Biotechnology 1996;14:620-623; Flores et al. Metabolic Engineering 2005; 7: 70-87). These previous reports were not considered in the present submission.

We thank the reviewer for the valuable comments. We have considered these issues in the revised manuscript, and based on these previous works developed a method, described in the text, to reproducibly grow \$\Delta ptsI\$ mutants on glucose. In short, by pre-culturing cells in M9 medium with galactose we induce the *gal* operon and this allows \$\Delta ptsI\$ mutants to grow on glucose. As a comparison, when cells are pre-cultured in LB medium, we do not observe any significant growth on glucose (see figure below). As an additional control, we have generated the \$\Delta ptsI \Delta glk\$ double knockout and demonstrated that this strain is unable to grow on glucose, even when pre-cultured on galactose. The new results have been added to the text. The suggested references and others have been added.

In lines 121-122, authors stated that during growth on xylose, EI mutation was non-essential for growth as in glucose condition. Additionally, in lines 138-139, was wrote that "The only non-growth phenotype observed was for $\Delta ptsI$ on glucose, as previously discussed." However, according to the previous reports, particularly data from Ling et al., EI mutants can grow in glucose at low growth rates. Based on this information, authors are prompted to consider additional experiments in glucose as carbon source. Results on xylose are valuable, of course, but demonstrate the role of the inactivation of EI in the back-flow of C from PYR→PEP growing in a non-PTS sugar.

Please see our response above. By pre-culturing cells culture with galactose, we are now able to conduct labeling experiments with $\Delta ptsI$ mutants grown on glucose. The manuscript has been updated accordingly. New results and discussion have been added to the text.

Regarding the experimental design with labeled alanine, authors are prompted to show the fate of the incorporation of this amino acid in the schematic representation of PEP synthesis in Figures 1-3 panel a, respectively.

The suggested change has been made.

As labeled alanine is incorporated to PEP available for cell growth, how much of the C from alanine contributed to the observed biomass in each experiment (acetate, glucose, and xylose)? Growth rates showed in Figure 1-3 panel b and Figure 4, panel b, were determined before the addition of the labeled bolus?

In all experiments we performed, no change in growth curves was observed upon addition of alanine tracers. Alanine is expected to equilibrate with pyruvate, but very little is expected to be catabolized and contribute net carbon to biomass formation. This is most clearly the case under glycolytic growth conditions, when carbon catabolite repression inhibits net alanine catabolism in most cases.

Minor comments:

Lines 22 and 41: Define the abbreviated form for pyruvate (PYR) as it is used in this from in the entire document. e.g. line 42: "PEP/Pyr". Use PYR instead pyruvate among the entire document.

The suggested change was made.

Lines 221, show the carbon sources and concentrations assayed. Why do you use different levels of each carbon source? Does this difference be related to the total moles of each C source? Moreover, differences influenced the total biomass obtained in each carbon source? Precise.

The amounts of C sources used in the media were based on preliminary results where our aim was to ensure that the primary C source would not be limiting during the culture period. The smaller inoculum and harvesting OD₆₀₀ for the acetate and pyruvate cultures were chosen to ensure that the alanine tracer remained in excess.

REVIEWERS' COMMENTS:

Reviewer #1 (Remarks to the Author):

The authors have revised and updated their manuscript and have included new experimental data in response to this referee's comments. The authors have also responded to referee 2 and have performed additional experiments which they present now.

From my view this has improved the manuscript and strengthened the claims of the authors.

The discussion is also rewritten and helps the reader better in understanding.

I have only one request: Bacterial enzymes should be abbreviated as, for example, PpsA and not as ppsA (gene nomenclature) throughout the manuscript.

For principal considerations, I prefer to stay anonymous a reviewer.

Reviewer #2 (Remarks to the Author):

The reviewed version of the manuscript entitled: Enzyme I facilitates reverse flux from pyruvate to phosphoenolpyruvate in *E. coli*, by Long et al., has attended all the criticisms and suggestions satisfactorily.

Authors provided additional experiments to answer some questions raised during the previous review. New results and discussion were added to the new version of the manuscript.

This current version of the manuscript is suitable for publication in Nature Communications.

Reviewer #1 (Remarks to the Author):

The authors have revised and updated their manuscript and have included new experimental data in response to this referee's comments. The authors have also responded to referee 2 and have performed additional experiments which they present now.

From my view this has improved the manuscript and strengthened the claims of the authors.

The discussion is also rewritten and helps the reader better in understanding.

We thank the reviewer for the positive comments.

I have only one request: Bacterial enzymes should be abbreviated as, for example, PpsA and not as ppsA (gene nomenclature) throughout the manuscript.

Corrected.

For principal considerations, I prefer to stay anonymous a reviewer.

Reviewer #2 (Remarks to the Author):

The reviewed version of the manuscript entitled: Enzyme I facilitates reverse flux from pyruvate to phosphoenolpyruvate in *E. coli*, by Long et al., has attended all the criticisms and suggestions satisfactorily.

Authors provided additional experiments to answer some questions raised during the previous review. New results and discussion were added to the new version of the manuscript.

This current version of the manuscript is suitable for publication in Nature Communications.

We thank the reviewer for the positive comments.